# Causal modelling of variation in clinical practice and long-term outcomes of ADHD using Norwegian registry data: the ADHD controversy project

Arnstein Mykletun [1,2,3,4] Tarjei Widding-Havneraas,[2,5] Ashmita Chaulagain,[2,5] Ingvild Lyhmann,[2,5] Ingvar Bjelland,[5,6] Anne Halmøy,[5,6] Felix Elwert,[7] Peter Butterworth,[8] Simen Markussen,[9] Henrik Daae Zachrisson,[10] Knut Rypdal[2]

For numbered affiliations see end of article.

**Correspondence to**
Professor Arnstein Mykletun;
arnstein.mykletun@uit.no

## ABSTRACT

**Introduction** Attention-deficit/hyperactivity disorder (ADHD) is among the most common mental disorders in children and adolescents, and it is a strong risk factor for several adverse psychosocial outcomes over the lifespan. There are large between-country and within-country variations in diagnosis and medication rates. Due to ethical and practical considerations, a few studies have examined the effects of receiving a diagnosis, and there is a lack of research on effects of medication on long-term outcomes. Our project has four aims organised in four work packages: (WP1) To examine the prognosis of ADHD (with and without medication) compared with patients with other psychiatric diagnoses, patients in contact with public sector child and adolescent psychiatric outpatient clinics (without diagnosis) and the general population; (WP2) Examine within-country variation in ADHD diagnoses and medication rates by clinics' catchment area; and(WP3) Identify causal effects of being diagnosed with ADHD and (WP4) ADHD medication on long-term outcomes.

**Method and analysis** Our project links several nationwide Norwegian registries. The patient sample is all persons aged 5–18 years that were in contact with public sector child and adolescent psychiatric outpatient clinics in 2009–2011. Our comparative analysis of prognosis will be based on survival analysis and mixed-effects models. Our analysis of variation will apply mixed-effects models and generalised linear models. We have two identification strategies for the effect of being diagnosed with ADHD and of receiving medication on long-term outcomes. Both strategies rely on using preference-based instrumental variables, which in our project are based on provider preferences for ADHD diagnosis and medication.

**Ethics and dissemination** The project is approved by the Regional Ethics Committee, Norway (REC number 2017/2150/REC south-east D). All papers will be published in open-access journals and results will be presented in national and international conferences.

**Trial registration numbers** ISRCTN11573246 and ISRCTN11891971.

## Strengths and limitations of this study

► This study will use reliable and complete national registry data to study real-world effects of attention-deficit/hyperactivity disorder (ADHD) diagnosis and medication on long-term psychosocial outcomes highly relevant for patients, families and society at large.

► We use observed variations in rates of ADHD diagnoses and medication between public sector child and adolescent psychiatric outpatient clinics for causal modelling, enabling empirical investigation of research questions which cannot be addressed in conventional randomised controlled trials for ethical and practical reasons.

► In Norway, ADHD in children and adolescents is almost exclusively diagnosed in public sector specialist services, which is covered by our registry data, ensuring representative data for the Norwegian general population.

► The observed variation in ADHD diagnosis and medication within Norway reflects most of the variation observed between countries, relevant to the external validity of the study.

► Whereas the strength of registry data as used in this study is related to data completeness and reliability, the weakness relates to validity and in some cases also data availability.

## INTRODUCTION

Attention-deficit/hyperactivity disorder (ADHD) is among the most common mental disorders in children and adolescents,[1–4] and is often associated with problems into adulthood.[5] ADHD is a persistent neurodevelopmental disorder characterised by symptoms of inattention and/or hyperactivity and impulsivity that are excessive for an individual's age or overall development.[6 7] ADHD is a strong risk factor for school drop-out,[8] injuries,[9] comorbidity,[10] self-harm and mortality,[11–14] contact with child welfare services,[15 16] as well as lack of employment,[17 18] later substance use disorders[17 19–21] and criminality.[22–24]

Clinical treatment guidelines recommend a multimodal approach that combine psychosocial and pharmacological treatment options.[25 26] Behavioural intervention such as parent training is presently considered an important part of the treatment for patients with ADHD.[27] While not proven to be effective in reducing core ADHD symptoms, evidence shows that it has positive effects on parenting and reduces oppositional and noncompliant behaviours in children, and may also lead to improvement in parent-child relationships as well as children's emotional, social and academic functioning.

In the case of pharmacological treatment, on the other hand, several clinical trials have provided evidence of psychostimulants' safety and efficacy in reducing ADHD symptoms in the short term.[28 29] Short-term studies show that medication reduces symptoms in at least 75% of children and adolescents with ADHD,[30] suggesting that medication may be superior to both psychological interventions and no treatment.[31] Evidence from a recent systematic review and network meta-analysis of 133 double-blind randomised controlled trials (RCTs) concluded that the first-choice medications for the short-term treatment of ADHD should be methylphenidate and amphetamines for children/adolescents and adults, respectively.[29]

While short-term effects of medication are established (although with some discussions surrounding the evidence base),[32] there are few studies on long-term effects of such treatment.[29] Further, what general conclusion to draw from the existing findings on this topic is still under discussion, as illustrated by a recent debate section in *Journal of the American Academy of Child & Adolescent Psychiatry*.[33–35] This debate illuminated that although several long-term studies indicate positive consequences of ADHD medication use, effect sizes are typically reduced compared with short-term results; additionally, in an even longer time perspective, it seems that the effectiveness of treatment-as-usual wanes and may not even be significant.[35] From the opposing standpoint of the debate, however, it is proposed that the apparent lack of long-term benefits of medication is likely caused by insufficient monitoring and adjustment of treatment for patients followed up over long periods of time, and should not be used as a general argument against the effectiveness of medication in the treatment of ADHD.[34]

A promising source of long-term effect estimates to contribute to this debate is nationwide register studies.[5 36 37] In recent years, register studies using quasi-experimental designs have shown that medication may reduce risk of unwanted outcomes associated with ADHD such as crime,[38–41] substance abuse,[42] injuries[9 43] and motor vehicle accidents.[44] However, as summarised in a recent review covering register-based studies on the effects of ADHD medication on a wide range of behavioural and neuropsychiatric outcomes, the available evidence even from register based pharmacoepidemiological studies on long-term effects of ADHD medication are still less clear than studies on short-term effects, and are subject to several limitations.[37] As pointed out by these authors and others, more long-term studies that examine functional outcomes in real-world settings are needed.[28 29 34 37 45]

Over recent decades, there have been substantial increases in clinical diagnosis and pharmacological treatment of ADHD in many countries,[2] including North America,[46] Europe,[47] Australia[48] and Asia.[49] Recent systematic reviews and meta-analyses estimate that the global prevalence rate of ADHD in children and adolescents is between 3.4%[1] and 7.2%.[50] It has been estimated that at least an additional 5% of children and adolescents have substantial difficulties with excessive and impairing levels of overactivity, inattention and impulsivity that are just below the threshold for diagnostic criteria.[51] Some studies attribute the variation in prevalence estimates to social and cultural factors,[52] while others downplay geographical location and argue for a methodological explanation.[3]

A retrospective study based on population-level databases from 13 countries and 154.5 million individuals showed that prevalence of ADHD medication use in children aged 3–18 years varied between 0.27% and 6.69% across included countries.[53] Among the Nordic countries, the prescription prevalence has been estimated to range from 0.12% in Finland to 1.25% in Iceland.[54] Further, large within-country variation in prescription rates has been reported in several countries, for example, the USA,[55–57] Denmark,[58 59] Sweden[60] and Germany.[61]

Even within Norway, a country with a modest inequality and a universal healthcare system, there are substantial regional differences in diagnosis and medication.[62 63] For instance, the countywise rates for ADHD diagnosis in Norway at ages 6–12 vary between 1.7 % and 4.8 % for boys and between 0.4 % and 2.0 % for girls, and similar variation has been noted for medication rates.[62] In total, 80% of the children and adolescents diagnosed with ADHD fill one prescription for ADHD medication,[62] and approximately 50% continue to collect prescriptions for more than 1 year.[64] In addition to geographical variation in diagnosis and medication rates, studies have shown that being young relative to peers is a strong predictor for being diagnosed with ADHD in several countries.[65 66] A Norwegian chart review indicated that only 49% of all diagnoses were adequately documented.[67]

Internationally, the variation in ADHD diagnosis and medication rates have caused a debate with two opposing perspectives. On the one hand, there is a restrictive perspective characterised by concerns about potential overdiagnosis of ADHD, leading to medicalisation of normal behaviour,[68] unnecessary stigma[69] and side effects of pharmacological treatment[70] that might not even be helpful in the presence of true ADHD. On the other hand, there is a liberal perspective distinguished by concerns about underdiagnosis and undertreatment,[71] which argues that increased rates of ADHD diagnosis and medication are due to improved recognition and understanding by professionals,[72 73] broadened diagnostic definitions[74] and methodological problems in the literature.[2] This perspective emphasises the importance of reducing

the risk of adverse outcomes associated with untreated ADHD, such as injuries,[9] school drop-out[8] and criminal behaviour.[43]

This debate may be reflected in clinical practice, where patients referred to specialist healthcare with suspicion of ADHD present with various degrees and compositions of symptoms (figure 1). Clinicians generally regard ADHD symptoms on a continuum. Although the diagnostic manuals define a symptomatic and functional cut-off for ADHD,[75] diagnosis is inevitably based on a clinical judgement. This discretionary decision is likely to vary between clinicians,[62] partly due to underlying restrictive or liberal perspectives. Figure 1 presents our theoretical model on clinicians' consensus on diagnosis and medication by ADHD symptom load among patients in contact with child and adolescent psychiatric outpatient clinics. Clinicians will generally agree that patients with very few or no symptoms of ADHD should receive neither a diagnosis nor medication (left side of figure 1).[76–78] On the opposite end of the continuum are patients with obvious ADHD symptoms, for whom most if not all clinicians will agree on ADHD diagnosis and medication (right side of figure 1).

The controversy on ADHD diagnosis and ADHD medication mostly relates to the middle area of the symptom distribution in figure 1, which includes patients with ambiguous symptoms, bordering on the threshold for diagnostic cut-off. These patients may be presenting with symptoms compatible also with other diagnoses or could be experiencing psychosocial stress that may produce ADHD-like symptoms, making an ADHD diagnosis difficult to establish.

In light of increasing ADHD diagnosis and medication rates in most Western countries, it is important to improve our knowledge on the effect of diagnosis among patients around the clinical threshold for diagnosis, as well as the effects of medication on long-term outcomes among patients. This is an important issue to address as it can inform the debate on underdiagnosis and overdiagnosis

and medication. A pressing issue is whether patients with ambiguous ADHD symptoms benefit or not from being diagnosed. As pointed out by Owens,[79] an ADHD diagnosis '… can bring beneficial pharmacological treatment and social supports, but it can also trigger negative social and psychological processes, as suggested by labelling theory'. Being diagnosed entails a set of at least three components that interact on long-term outcomes: (1) stigma ('diagnosis labelling effects'),[76 80–83] (2) medication and (3) psychosocial treatment. Owens[79] finds that consistent with labelling theory (stigma), only those with severe prediagnosis symptoms seem to benefit from diagnosis and medication, while those with mild prediagnosis symptoms have worse outcomes compared with undiagnosed matches.

The key issue in identifying an effect of receiving a diagnosis is that diagnoses cannot be randomised to patients. Hence, we must either rely on epidemiological designs that precludes causal inference unless all potential confounding is statistically controlled, or we need to find sources of plausibly exogenous variation that can be used as a natural experiment. We believe the geographical variation in diagnosis and medication can be used as a natural experiment to identify causal effects of diagnosis and medication on long-term outcomes. There are several plausible explanations for geographical variation in diagnosis and medication rates on both the supply and demand side, including resources, capacity and patient mix. Nevertheless, a portion of the variation may be due to variation in clinician's diagnosis and medication practice among public sector child and adolescent psychiatric outpatient clinics, commonly referred to as provider preference,[84] which can lead to variation in diagnosis and prescription decisions for similar patients. Geographical variation may induce some random variation in treatment status for patients and may be used as an instrumental variable (IV) to obtain treatment effects.[85] Considering the regional differences in ADHD diagnosis and treatment, we expect that the child and adolescent outpatient clinics in Norway have different preferences. In single-provider universal access health systems, variation in health service provision that is not accounted for by differences in patient populations and symptom load is usually regarded as unwarranted and unwanted.[86]

The current research project aims to contribute to the existing knowledge on ADHD diagnosis and ADHD medication in several ways. Our project will employ Norwegian registry data to explore four main aims with corresponding work packages (WP):

WP1: The epidemiology of ADHD prognosis. The main aim of this WP is to describe the general long-term prognosis of ADHD compared with the general population. This is in response to the typical parental concern: 'My child is diagnosed with ADHD, how will he/she do in life?' By long term, we here mean years rather than months, including the important transition from childhood and adolescence to early adult life. We will examine the prognosis of patients diagnosed with ADHD in child

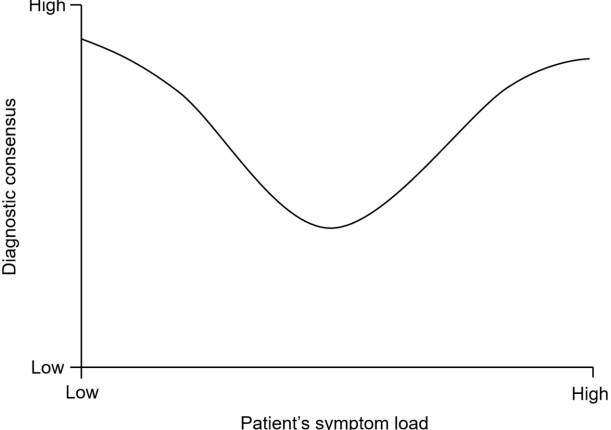

**Figure 1** Theoretical model on clinicians' consensus on diagnosis and medication by ADHD symptom load among referred patient. ADHD, attention-deficit/hyperactivity disorder.

and adolescent psychiatry (with and without medication). These will be compared with patients in child and adolescent psychiatry who received other diagnoses, or who received no diagnosis. We will also compare to the general population without contact with child and adolescent psychiatry. Measures of prognosis include registry data records of emergency care visits/injuries, comorbidity, contact with child welfare services, contact with adult psychiatric health services, education, employment and income, welfare dependency, crime, self-harm, suicide and mortality.

WP2: The ADHD variation controversy. This WP has two aims: first, we will describe the geographical variation in ADHD diagnosis and medication in Norway. We will describe this variation at the level of the catchment areas for child and adolescent psychiatric outpatient clinics in Norway. We will describe the variation in ADHD diagnosis as proportion of children and adolescents referred to these clinics and as proportion of children and adolescents in the catchment area, and also in ADHD medication as proportion of diagnosed patients. Second, we aim to explain this variation by three strategies:

1. We will explore to what extent this variation in clinical practice can be attributed to catchment area characteristics. Data on catchment areas will also be included in WP3 and WP4 as potential confounding variables.
2. Further, we will explore to what extent it can be attributed to the variation in observed ADHD symptoms as observed in a nationwide health study of the general population.[87] At an aggregated level, a strong positive association between the observed symptoms (as self-reported by mothers) and rates of diagnoses and medication (as observed in registry data) will challenge the instrument planned for this study.
3. Finally, we will survey clinicians working at the about 90 child and adolescent psychiatric outpatient clinics in Norway, exploring the variation in their attitudes and opinions towards ADHD diagnosis and medication. Specifically, we will explore if these attitudes and opinions vary on a continuum from restrictive to liberal. We will also explore if this variation in attitudes and opinions correlate at the aggregated level of the clinics with observed rates of ADHD diagnosis and medication (according to the registry data), and with the variation in ADHD symptoms (according to the health study[87]).

WP3: Long-term effects of receiving an ADHD diagnosis. The main aim of this WP is to examine if it makes any long-term difference for the prognosis of patients if they attend clinics with liberal or a restrictive practice on ADHD diagnosing. We assume that clear ADHD will be diagnosed at all clinics, and that clear non-ADHD patient will not receive the ADHD-diagnosis anywhere. Consequently, this WP aims to analyse the effect of ADHD diagnosis for patients on the margin of receiving the diagnosis, who are most likely to be treated differently at clinics with a liberal vs restrictive approach to the diagnosis. We will identify causal effects of being diagnosed with ADHD

among patients in contact with child and adolescent psychiatric outpatient clinics on long-term outcomes including emergency care visits/injuries, comorbidity, education, contact with child welfare services, contact with adult psychiatric health services, employment and income, welfare dependency, crime, self-harm, suicide and mortality.

WP4: Long-term effects of ADHD medication. The main aim of this WP is to examine if it makes any long-term difference for the prognosis of patients diagnosed with ADHD whether they attend a clinic with a liberal or restrictive ADHD medication practice. Not all patients who receive the ADHD diagnosis will be prescribed medication, and we assume that patients who are on the margin of receiving such treatment will be treated differently at clinics with a liberal versus restrictive approach. We will identify causal effects of initial ADHD medication among patients diagnosed with ADHD on long-term outcomes including emergency care visits/injuries, comorbidity, education, contact with child welfare services, contact with adult psychiatric health services, employment and income, welfare dependency, crime, self-harm, suicide and mortality.

Registry-based studies on effects of ADHD-medication that address limitations of RCTs, such as generalisability, follow-up time and less frequent psychosocial outcomes, is expanding.[37] The main challenge of these studies is confounding, which several studies try to address by research designs such as within-subjects designs,[37] difference in differences[9] and IVs.[43 88] Our project's original contribution lies in using two identification strategies which have not previously been applied to evaluate effects of diagnosis nor medication on several included outcomes. There has only been conducted one IV analysis of ADHD medication effects on hospital contacts and criminal behaviour using Danish nationwide registry data and one IV study on ADHD medication effects on teens risk of sexual behaviour outcomes, substance abuse and injuries using Medicaid data.[88]

## METHODS AND ANALYSIS
### Project period
The project is funded by a grant from the Western Norway Regional Health Authority starting 1 January 2018 lasting 6 years, and a grant from the Research Council of Norway lasting 4 years starting 1 November 2019. The project period may be extended with additional funding and for personnel-related reasons.

### Design
Our project employs methods for descriptive analyses and causal inference.[89 90] Our study draws on several linked Norwegian nationwide registries. Nordic registry data provide individual level data for the whole population,[91] and are only censored by migration and death.[92] Our project has four WPs:

## WP1: the epidemiology of ADHD prognosis

This first work-package is conventional epidemiology, analysing outcomes (prognosis) in the ADHD population compared with the general population. The aim is to provide general data on the prognosis of the ADHD population, employing all available outcomes. We will use survival analysis[93] to examine time to event for our outcomes. In line with recent studies,[94] we will account for possible competing risks to avoid overestimation of the cumulative incidence of our events of interest. We will also apply mixed-effects analysis, which is well suited and an established standard for individual-level longitudinal data with repeated observations.[93 95]

## WP2: the ADHD variation controversy

This second WP will employ conventional statistics including scatter plots and regression models. Most analyses will be performed on data aggregated to geographical regions (counties or finer) and (when data allows for this) to the about 90 child and adolescent psychiatric outpatient clinic catchment areas. To account for patient clustering and repeated observations over time, we will use standard generalised linear models with cluster-robust SEs and mixed-effects models.[93 96]

## WP3 and WP4: long-term effects of receiving an ADHD diagnosis and ADHD medication

To identify the effect of ADHD diagnosis (WP3) and ADHD medication (WP4) on long-term outcomes, we will use child and adolescent psychiatric outpatient clinics' catchment area variation in diagnosis and medication as IVs. IV is a common identification strategy in health services research[97] that exploits the random variation induced in a treatment variable (here, ADHD diagnosis or ADHD medication prescription) by a plausibly exogenous (as-if random) event or process (ie, the instrument). The role of the instrument is to 'isolate' exogenous variation in the treatment, and hence to remove the endogenous (confounded) variation, so that the analysis only uses the exogenous variation to estimate the treatment effect.[98] In this sense, IV analysis is a quasi-experimental approach. There is evidence to support that research designs with valid IVs can provide similar estimates as randomised experiments on the same research questions,[97 99 100]

whereas more traditional observational studies without a credible source of exogenous variation often do not.[98 100] Our identification strategies exploit as-if random variation in providers' preference to diagnose patients with ADHD and to prescribe ADHD medication[43 88] across Norwegian public sector child and adolescent psychiatric outpatient clinics. The main idea behind provider preference-based instruments is that '…different providers or groups of providers have different preferences dictating how medications or medical procedures are used.'[84] If the provider preference is associated with treatment, does not directly affect the outcome, and is not associated with unmeasured risk factors of the outcome, it may be used as an IV to consistently estimate the treatment effect.[85] We will use two candidate provider preference instruments, illustrated in figure 2.

## Preference-based IVs

It is plausible that a provider's preference for diagnosing ADHD affects whether or not an individual is diagnosed with ADHD, while it is not associated with the outcomes of interest (ie, via paths other than affecting diagnosis) net of certain controls. We will use provider preference for ADHD diagnosis to identify the effect of diagnosis on long-term outcomes. Similarly, provider preference for ADHD medication likely affects whether or not a patient receives medication, while it is not associated with our outcomes of interest, net of certain controls. Provider preference for ADHD medication will be used to identify the effect of ADHD medication on long-term outcomes. Provider preferences for medication have become a well-established IV in epidemiology,[101] as they are usually strongly associated with medication use and reduce covariate imbalance.[102] Our IVs will be defined at the provider level (ie, the catchment area of child and adolescent psychiatric outpatient clinics), which may be stronger and more efficient compared with physician-level instruments.[103] The (catchment) area definition of child and adolescent psychiatric outpatient clinics strikes a reasonable balance between methodological issues introduced by too large or too small area definitions.[104]

To our knowledge, provider-level preference for diagnosis[62 63] is a novel candidate instrument, while provider

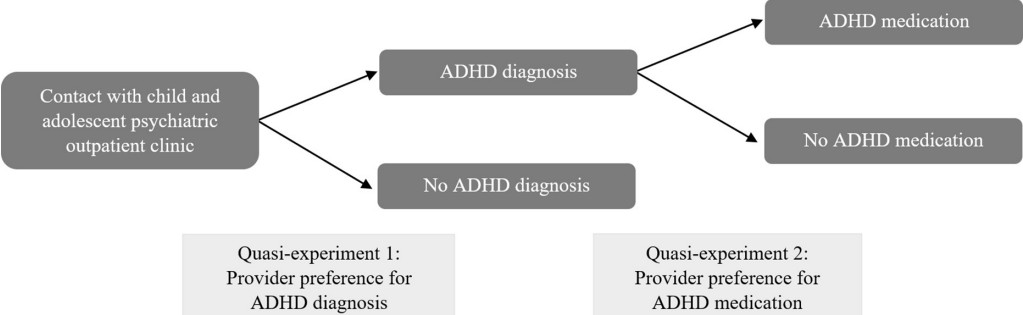

**Figure 2** Two sources of as-if randomisation. Quasi-experiment 1 is provider preference for ADHD diagnosis. Quasi-experiment 2 is provider preference for ADHD medication among those diagnosed with ADHD. ADHD, attention-deficit/hyperactivity disorder.

preference for ADHD medication has only been used in at least four prior studies,[43 88 105 106] which do not cover all outcomes in our project. We illustrate the variation that will be used as provider preference instruments for the effects of diagnosis (WP3) and medication (WP4) at the county level in figure 3. Our eventual analysis will use preferences measured at the level of catchment areas of child and adolescent psychiatric outpatient clinics.

Although rates of diagnosis and medication follow each other closely, preliminary evidence on the county level indicates that they do not correlate perfectly, as shown in figures 3 and 4.

### Setting of study and assessment of identification strategy

Preference-based instruments may lead to biased inferences if the identifying assumptions are not met.[107] The application of IV rests on a set of key assumptions that must be met to warrant a credible causal interpretation of estimates. Overall, an IV-strategy can be evaluated on a continuum from low to high credibility based on the underlying assumptions.[108] The two main assumptions of IV are relevance and exclusion. Relevance can be tested statistically, while exclusion can only be partially tested and relies on justification derived from subject-matter knowledge. The instruments are likely to meet the relevance assumptions as public sector child and adolescent psychiatric outpatient clinics are the only institutions in Norway that can diagnose ADHD and initiate pharmacological treatment of ADHD (aside from a small number of private practices), and the share of patients diagnosed

with ADHD and medicated for ADHD varies considerably across areas.

We expect that the instrument meets the exclusion restriction in several respects. We expect provider preference for diagnosis and medication only to affect long-term outcomes through diagnosis and treatment decisions. Although patients in Norway are free to choose outpatient clinics, which opens the door to self-selection bias in principle, in practice, the vast majority of patients use the nearest child and adolescent psychiatric outpatient clinic for convenience and because child and adolescent psychiatric outpatient clinics are expected to deliver equal services of equal quality.[109] There is no competitive market in this part of the health sector. Furthermore, it is unlikely that patients would sort in response to provider preferences, as the variation in rates of ADHD diagnosis and treatment are not common knowledge among patients or their parents. Consequently, children with similar ADHD symptoms are *as-if* randomly assigned to ADHD medication or no ADHD medication at a national level.

Next, we will control for the characteristics of the catchment area and the patient mix of each provider that may partially explain variation in diagnosis and prescription practices. We have relied on existing research and expert knowledge in our assessment of relevant control variables and included variables in our data accordingly.[110] In addition to our microlevel data, we construct a municipality-level data set from publicly available register data to further

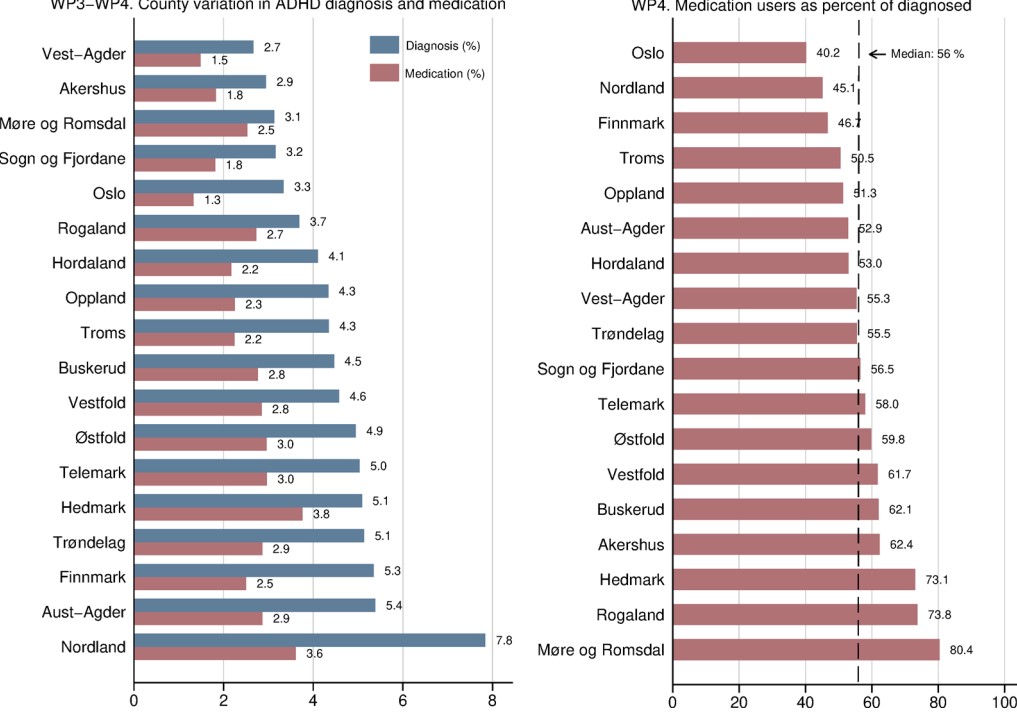

**Figure 3** County variation in ADHD diagnosis and medication percent for children and adolescents between 10 and 19 years in Norway. ADHD as main diagnosis registered once or more times during 2009–2011. Registered diagnosis in NPR 2009–2011 and registered filled ADHD prescription NorPD 2012. ADHD, attention-deficit/hyperactivity disorder; NorPD, Norwegian Prescription Database; NPR, Norwegian Patient Register; WP, work package.

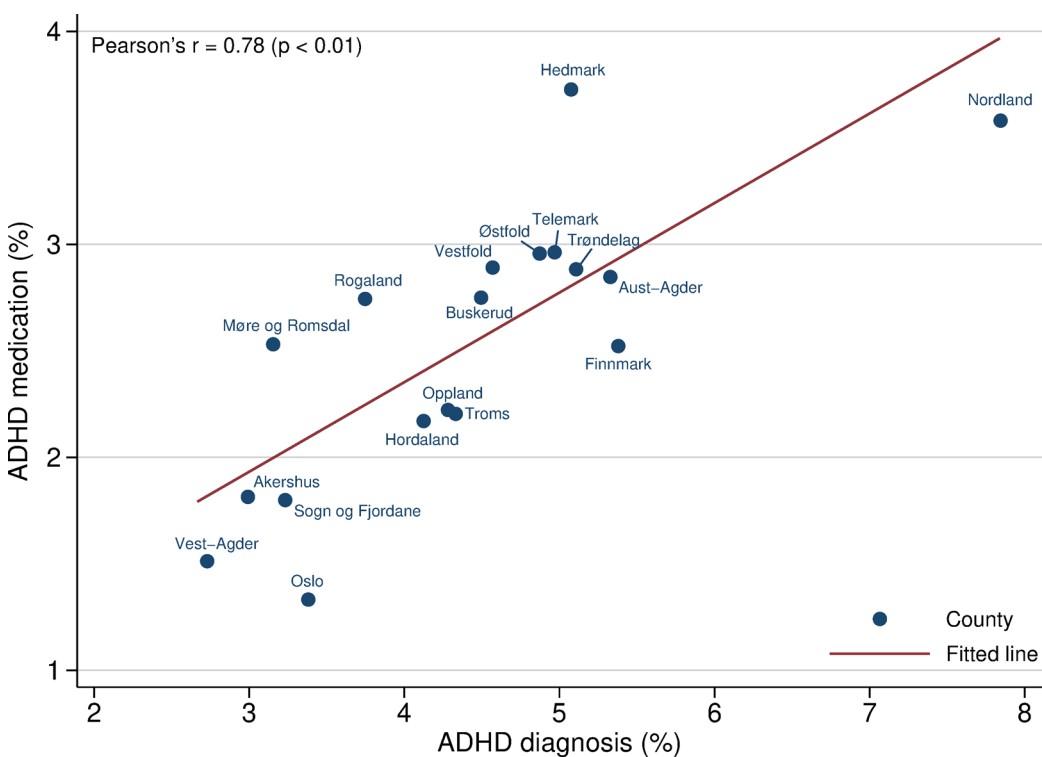

**Figure 4** Scatterplot of ADHD diagnosis and medication among children and adolescents between 10 and 19 years by Norwegian counties. ADHD as main diagnosis registered once or more times during 2009–2011. Linear fitted line. Registered diagnosis in NPR 2009–2011 and registered filled ADHD prescription NorPD 2012. ADHD, attention-deficit/hyperactivity disorder; NorPD, Norwegian Prescription Database; NPR, Norwegian Patient Register.

improve our adjustment for potential confounding. (Our list of potential confounders is presented in table 1). Exclusion is plausible due to the existence of large geographical variation in ADHD diagnosis and medication, net of demographic factors,[43 62 111] which suggests that clinical practice varies for idiosyncratic reasons. As further validity assessment, we will analyse whether there is geographical variation in ADHD symptoms using the nationwide The Norwegian Mother and Child Cohort study (MoBa), and whether referral frequencies vary geographically using data from the Norwegian Patient Register.

| Table 1 | Potential confounders for facility-level instrumental variables | | |
|---|---|---|---|
| **Confounder category** | **Potential confounders** | **Covered** | **Data source** |
| Geographic | Urban/rural | Yes. | Municipality data set and municipality level covariates in microdata (table 2). |
| Patient characteristics | Race, education, income, age, insurance status, health status/comorbid conditions, health behaviours | Insurance status is not likely an issue in the Norwegian healthcare system. Health behaviour is unobserved. | Microdata (table 2). |
| Facility characteristics | Procedure volume, facility volume, clinical services offered, departments, teaching status, profit status, trauma designation, delivery system type, practice type | Although the sizes of the facilities vary, the provided services and the organisation of the facility are part of the same nationwide healthcare system and should be similar. Profit status is likely not an issue in Norwegian healthcare system. | Municipality data set and municipality level covariates in microdata (table 2). |
| Treatment characteristics | Receipt of other treatments | We may adjust for contacts with health services for treatment and common prescriptions (eg, melatonin, antidepressants) | Microdata (table 2). |

Table based on Garabedian et al.[110]

## Triangulation and statistical methodology

Our assessment of causal effects relies on IV triangulated with other methods based on other assumptions.[112] As pointed out by Matthay *et al*.[113]: 'All methods involve untestable assumptions and trade-offs in precision and validity' and there is seldom a '… clear justification for exclusive reliance on one method'. We will supplement our IV-analysis with mixed-effects models, propensity score matching, inverse-probability weighting and fixed-effects regression (for medication).[96] These are also the methods we will base our analysis on if IV-conditions are not met. All analyses will be performed in Stata 16.1[114] and R.[115]

## Data and study sample

Our data comprises one sample of patients and one sample of controls from the general population. The patient sample includes all persons between 5 and 18 years of age that had any contact (for any reason) with child and adolescent psychiatric outpatient clinics between 2009 and 2011 (ie, birth cohorts 1991–2006). Children younger than 5 years are not included as few children are diagnosed with ADHD before age 5.[43] The control sample is a random sample from the general population of equal size as the patient sample defined as individuals in Norway 5–18 years old without reported contact (for any reason) with child and adolescent psychiatric outpatient clinics in 2009–2011. The control sample is included to enable us to compare the prognosis among patients diagnosed with ADHD and patients in contact with child and adolescent psychiatric outpatient clinics to the general population.

As the project will examine outcomes that occur at different points over a life course (injuries, contact with child welfare services, contact with adult psychiatric health services, education, income, welfare dependency, comorbidity, crime, self-harm, suicide and mortality), we will receive data for 2009–2019 in 2020 and for 2020–2021 in 2022. We have estimated that the patient sample with an equally sized control sample from the general population (matched on age, gender and geography) will be approximately 200 000. All analyses are registry based. We combine data from a total of nine national registries, presented in table 2.

The registry data set will be supplemented with a municipality data set based on publicly available aggregated administrative data. The municipalities will be grouped by child and adolescent psychiatric outpatient clinic catchment areas. The municipality data set serves two purposes: (1) examine associations between environmental factors and geographical variation in ADHD diagnosis and prescription patterns and (2) account for population mix at child and adolescent psychiatric outpatient clinics in analyses. The municipality data set

| Table 2 | Data from registries included in the project |
|---|---|
| **National register** | **Variables** |
| Norwegian Patient Register | Diagnoses (International Statistical Classification of Diseases and Related HealthProblems versjon 10 (ICD-10), all F-diagnoses by multiaxial coding), institution, in/out date, waiting time code, contact type, referral from institution, referral type, referral reason, injury (specialist care), injury severity, child welfare service involvement, child's care situation, child's environment, gender, age, patient catchment area |
| Control and Payment of Health Reimbursement Register | Injuries (emergency room) International Classification of Primary Care, Second edition (ICPC-2) diagnoses (somatic: fracture, concussion, eye injuries, penetration, burn, poisoning, other injuries/ psychiatric: suicide attempt, hyperkinetic disorder, anorexia, schizophrenia) |
| Norwegian Cause of Death Registry | Death, cause of death (three categories: suicide, injury, other) |
| Norwegian Prescription Database | Psychostimulants for ADHD (Anatomical Therapeutic Chemical (ATC)-codes N06B, CO2, CO2A), melatonin (ATC-codes N05, N05B), antidepressants (ATC-code N06A), antiepileptics (ATC-code N03A), defined daily dose |
| The Central Penal and Police Register (Strasak) | Criminal charges (ten categories: property theft, other offences for profit, criminal damage, violence and maltreatment, sexual offences, drug and alcohol offences, public order and integrity violations, traffic offences, other offences) |
| The Central Population Register | Country of origin (world region), parents' marital status, emigration |
| Municipality-State-Reporting | Case type, who reported case, content of case, measure in child welfare services law, removal of custody, reason for administrative decision |
| Norwegian Education Database | Highest completed level of education, parents' educational level, ended high school education before completion ('dropout'), average grade level for primary school, grades from high school completion, national test scores, order/conduct, absence |
| Income, tax and wealth register | Income, work assessment allowance, social assistance, disability pension, unemployment assistance ('dagpenger'), parents' income |

ADHD, attention-deficit/hyperactivity disorder.

will include covariates for socioeconomic conditions (education, income, welfare recipiency), crime, income inequality, living conditions, health service utilisation and health service economic variables.

### Patient and public involvement

Patients and public were not involved in the development of WPs or study design.

### DISCUSSION

This project examines (1) the prognosis of ADHD, (2) within-country variation in ADHD diagnosis and medication rates by clinics' catchment area, (3) causal effects of being diagnosed with ADHD on long-term outcomes and (4) causal effects of ADHD medication on long-term outcomes. The project relies on Norwegian nationwide register data and employs several methodological approaches, including survival analysis, mixed-effects models and IVs.

As outlined in the introduction, the debate concerning the long-term consequences of variation in diagnosis and medication of ADHD is far from settled. RCTs on ADHD may not adequately address the controversy on long-term effects of ADHD diagnosis or medication, as randomisation of these exposures on a large scale with long follow-up is infeasible. Causal inference from conventional epidemiological designs is also generally implausible, due to unobserved confounding. By combining rich nationwide register data with an IVs design, we are able to circumvent the shortcomings of both RCTs and conventional epidemiological designs on these topics. Our approach aims to fill this knowledge gap by analysing effects of diagnosis and medication for children and adolescents on long-term outcomes.

In this sense, this project has merit to provide empirical evidence that neither RCTs nor conventional epidemiological designs can address. First, there is to our knowledge no trial randomising children to ADHD diagnosis. Our approach aims to fill this knowledge gap by instrument variable analysis for children at the margin of an ADHD diagnosis. Second, most clinical trials of medication in ADHD to date has been conducted on children that clearly meet diagnostic criteria, whereas the variation and controversy is regarding children on the margin of receiving medication.

This project is the first detailed examination of within-country variation in ADHD diagnosis and ADHD medication by child and adolescent outpatient psychiatric clinics' catchment areas, including possible associations with socioeconomic and demographic covariates. The project will be the first application of provider preference for ADHD diagnosis as an IV. The project will also be the first application of IVs to assess the effects of ADHD diagnosis and ADHD medication on several psychosocial outcomes.

The project is subject to the following limitations: Register data are limited to precollected data in registers, which also determine data selection and data quality.[116] Register data can be considered as the 'administrator's view of the world' which affects data definitions and data collection practice, and often makes it challenging to understand data-generating processes.[116] Moreover, register data are generally considered reliable, but the validity is important to assess due to complex data collection processes that varies between registers.[92 117]

### ETHICS AND DISSEMINATION
#### Data management

The Norwegian Prescription Database (NorPD) will combine data from the different registries and replace the personal identification number with a project-specific number (pseudonymisation) before distributing data to the research group to protect the privacy of participants. NorPD will keep the linkage key between the personal identification number and the project-specific number. Data will be stored and analysed on a secure server at Haukeland University Hospital that only the PI and other preapproved researchers will have access to. To ensure the quality and reproducibility of our data management and statistical analyses, we will (1) cross-check each step in the data cleaning and preparation process and (2) document each step from the raw data to the final data sets and statistical analyses in version-controlled syntax and log-files. Missing data will be assessed, and we will consider strategies for multiple imputation.

#### Ethics approval and consent to participate

The project is approved by the Regional Committee for Medical and Health Research Ethics, Norway: REC South East, Committee D (REC number 2017/2150/REC southeast D). Due to the large number of participants, it is not feasible to obtain consent from all participants. The study has been evaluated by REC to be 'of considerable importance to society' and to 'protect the integrity and welfare of the participants'. Based on the fulfilment of these criteria, the project has been exempted from the requirement of collecting informed consent from participants. The REC approval is given under the precondition that the project is conducted as described in the project application, protocol, communication and the administrative provisions in the Norwegian Health Research Act ('Helseforskningsloven') and appurtenant regulations.

#### Availability of data and materials

The data used in this project may not be made publicly available due to the sensitive nature of the data. Current approvals and regulations do not allow sharing these data. Due to general data protection regulations, restrictions in the ethics approval and restrictions by the registry owners, data may not be shared. We may share statistical code.

#### Dissemination

The publication strategy distinguishes between six types of publications planned from this project:
1. Preparatory publications: This includes the present protocol paper, trial registrations, systematic literature

reviews on subject matter issues regarding ADHD[118] and provider preference IV applications.[119]

2. WP1: The epidemiology of ADHD prognosis. We will utilise all data to describe common trajectories in ADHD. Multiple comparison groups will be employed including other patients referred to child and adolescent psychiatry who got other diagnoses or no diagnosis, and also the general population. Outcomes are according to data as described in table 2.

3. WP2: The variation in ADHD diagnosis and treatment. We will describe in detail the geographical variation in ADHD diagnosis and medication, and also explore if characteristics of the catchment areas may explain some of the observed variation in clinical practice. We will publish results from the survey among clinicians at the public sector child and adolescent psychiatric outpatient clinics in Norway, exploring if their opinions and attitudes to ADHD diagnosis and treatment vary from restrictive to liberal, and if this variation corresponds with the observed geographical variation in rates of ADHD diagnosis and medication. Further, we will publish results from the health survey, exploring if there is an association between rates of ADHD symptoms and ADHD diagnosis and medication.

4. WP3: Long-term effects of receiving an ADHD diagnosis for children on the margin of the diagnostic threshold. Publications on the effect of ADHD diagnosis among those referred to child and adolescent psychiatry on outcomes as described above. Outcomes according to data in table 2.

5. WP4: Long-term effects of ADHD medication. Publications on the effect of ADHD medication among those diagnosed with ADHD on outcomes as described above. Outcomes according to data in table 2.

6. Exploratory publications on hypotheses and topics beyond those here described. These may be subject to additional approvals from the ethics committee and the owners of the registries, eventually also an update of the Data Protection Impact Assessment.

We will follow International Committee of Medical Journal Editors Vancouver recommendations for authorship when publishing the papers. Most of our team members have significant experience in writing and publishing scientific papers, and we believe our multidisciplinary team's experience will serve as the main expertise in writing the planned articles. We aim to publish with open access. Substantial amendments to our protocol will be reported to relevant parties through relevant channels.

**Author affiliations**
[1]Department of Community Medicine, University of Tromso Faculty of Health Sciences, Tromso, Norway
[2]Centre for Research and Education in Forensic Psychiatry, Haukeland University Hospital, Bergen, Norway
[3]Centre for Work and Mental Health, Nordland Hospital, Bodø, Norway
[4]Division for Health Services, Norwegian Institute of Public Health, Oslo, Norway
[5]Department of Clinical Medicine, University of Bergen, Bergen, Norway
[6]Division of Psychiatry, Haukeland University Hospital, Bergen, Norway
[7]Department of Sociology, University of Wisconsin-Madison, Madison, Wisconsin, USA
[8]Research School of Population Health, The Australian National University, Canberra, Victoria, Australia
[9]Ragnar Frisch Centre for Economic Research, Oslo, Norway
[10]Department of Special Education, University of Oslo, Oslo, Norway

**Acknowledgements** Several academic colleagues have advised on the development of the empirical and theoretical approach for this project in the context of the broader project 'Current Controversies in Psychiatry'. We acknowledge (in alphabetical order) Stål Bjørkly, Centre for Research and Education in Forensic Psychiatry, Oslo University Hospital, Oslo, Norway; Beate Brinchmann, Nordland Hospital Trust, Centre for Work and Mental Health, Bodø, Norway; Ian Colman, University of Ottawa, Canada; Sarah Dorrington, Institute of Psychiatry, Psychology and Neuroscience, King's College London, England; Kevin Douglas, Department of Psychology, Simon Fraser University, Canada; Simon Gilbody, University of York, York, England; Rolf Gjestad, Center for Research and Education in Forensic Psychiatry, Haukeland University Hospital, Bergen, Norway; Nicholas Glozier, Brain and Mind Research Institute, University of Sydney, Sydney, Australia; Stephen Hart, Department of Psychology, Simon Fraser University, Canada; Samuel Harvey, School of Psychiatry, University of New South Wales, Sydney, Australia; Hallvard Lund-Heimark, NORMENT Centre of Excellence, Haukeland University Hospital, Bergen, Norway; Matthew Hotopf, Institute of Psychiatry, Psychology and Neuroscience, King's College London, England; Erik Johnsen, Division of Mental Health and NORMENT Centre of Excellence, Haukeland University Hospital, Bergen, Norway; Eoin Killackey, Orygen & Centre for Youth Mental Health, University of Melbourne, Australia; Caroline Logan, National Health Service, England; Liv Mellesdal, Division of Mental Health, Haukeland University Hospital, Bergen, Norway; Tom Palmstierna, Department of Clinical Neuroscience, Karolinska Institute, Stockholm, Sweden; Mette Senneseth, Center for Research and Education in Forensic Psychiatry, Haukeland University Hospital, Bergen, Norway; Rob Stewart, Institute of Psychiatry, Psychology and Neuroscience, King's College London, England; for valuable advice and contributions in the process of developing this project in the broader framework 'Current controversies in psychiatry'.

**Contributors** AM proposed the idea and developed it in collaboration with TW-H and KR. AM is PI for the project and responsible for ethics, funding, main supervision and data management. TW-H, IL, and AC are PhD candidates, TW-H with a dual role also as statistician. FE, PB, SM and HDZ are involved for their expertise on statistics, epidemiology, and IV-methods, and have all advised on this project. FE developed the IV-strategy in collaboration with co-authors. IB and AH advised on the clinical substance matter and on the organisation of ADHD treatment delivery in Norway. KR is head of the unit where the study belongs. AM planned the structure of this protocol paper, and the first draft was written in close collaboration between TW-H, AC and IL. All other collaborators have contributed to the intellectual content of the paper, edited and approved the final manuscript.

**Funding** The project is funded by the Research Council of Norway (RCN) under the FRIMEDBIO (project number 288585) for four years starting 01.11.2019 and a grant from the Western Norway Regional Health Authority (project number 912197) from 1 January 2018 to 31 December 2023. (The project period and the funding will be extended due to delays related to the COVID-19 situation and other reasons for delay.) These are governmental funding bodies and the project has undergone peer-review as part of the funding application processes. The project is owned by Centre for Research and Education in Forensic Psychiatry, Haukeland University Hospital, Bergen, Norway, where the PI and PhD candidates on the project are currently employed.

**Disclaimer** The funders had no role in study design, data collection and analysis, decision to publish, or preparation of the manuscript.

**Competing interests** None declared.

**Patient and public involvement** Patients and/or the public were not involved in the design, or conduct, or reporting, or dissemination plans of this research.

**Patient consent for publication** Not required.

**Provenance and peer review** Not commissioned; externally peer reviewed.

ORCID iD
Arnstein Mykletun http://orcid.org/0000-0002-3878-0079

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
