## [Reviewer comments · BMJ Open]

ARTICLE DETAILS

TITLE (PROVISIONAL)	Causal Modeling of Variation in Clinical Practice and Long-Term Outcomes of ADHD using Norwegian Registry Data: The ADHD Controversy Project
AUTHORS	Mykletun, Arnstein; Widding-Havneraas, Tarjei; Chaulagain, Ashmita; Lyhmann, Ingvild; Bjelland, Ingvar; Halmøy, Anne; Elwert, Felix; Butterworth, Peter; Markussen, Simen; Zachrisson, Henrik; Rypdal, Knut

VERSION 1 – REVIEW

REVIEWER	Samuele Cortese University of Southampton
REVIEW RETURNED	11-Aug-2020

GENERAL COMMENTS	This is a large and ambitious, but much needed project, focusing on the key controversy/needs in the field of ADHD research. The approach is methodologically sound and I appreciate the fact that the authors seem to take a balanced view, rather than a biased one. The proposed methods are, in my view, solid and the manuscript is generally well presented, but , given the complexity of the aims, it could benefit from additional clarity More specifically, it would be important to summarise/highlight, after the initial discretion of each WP, what it aims to add to current literature (which are the gaps each WP aims to fill) Also, I'd suggest to provide a more "evidence-based, updated" introduction, rather than citing an old and, to some extent, biased (after the initial RCT phase) study as the MTA. the author may want to cite recent evidence synthesis on ADHD medications effects in the short term (eg: https://pubmed.ncbi.nlm.nih.gov/30097390/) and on the pharmacological strategies- in particular behavioural (eg: https://pubmed.ncbi.nlm.nih.gov/29083042/) as WP4 is focused on the issue of the long term, the authors may want to cite and discuss, in the discussion section, the main points in a controversy paper in JAACAP (https://pubmed.ncbi.nlm.nih.gov/31254608/, https://pubmed.ncbi.nlm.nih.gov/31515164/, https://pubmed.ncbi.nlm.nih.gov/31515165/), highlighting how their project addresses them Also, as the field lacks a standard definition of "long-term", the authors should clarify what "long-term" refer to in their project It would also be good to highlight also what their project adds to the large number of studies form the Swedish, USA, and Hong Kong registries (https://pubmed.ncbi.nlm.nih.gov/31155139/_)
--

REVIEWER	Peik Gustafsson Lund University, Medical Faculty, Department of Clinical Sciences Lund, Child and Adolescent Psychiatry. Lund, Sweden
REVIEW RETURNED	24-Sep-2020

GENERAL COMMENTS	Your planned project is very interesting. I have only one concern: You will base your study on differences in provider preference. This is an interesting approach as a kind of quasi-experimental design. The problem with this design is that there may be some differences in the true prevalence of ADHD and in referral frequencies in different geographical areas, not just reflecting provider preference. The problem will be to analyse relevant potential confounding and compensate for it, which might be difficult. Examples of confounding could be that some people might move to a university city for studies and academic career and that fewer parents of children with ADHD will do this (many of these parents have ADHD themselves). Special schools and better chance of getting a job could cause individuals (parents and children) with ADHD to move into a certain area. Different geographical areas might have different knowledge of and different attitudes towards the ADHD diagnosis, causing differences in referral frequency not reflecting the provider preference. I would appreciate if you could discuss these problems in greater detail and make a list of such potential confounders and to what degree you could compensate for them for in your study.
--

VERSION 1 – AUTHOR RESPONSE

Reviewer: 1

Reviewer Name: Samuele Cortese

Institution and Country: University of Southampton Competing interests: None

Comments to the Author

This is a large and ambitious, but much needed project, focusing on the key controversy/needs in the field of ADHD research. The approach is methodologically sound and I appreciate the fact that the authors seem to take a balanced view, rather than a biased one.

The proposed methods are, in my view, solid and the manuscript is generally well presented, but , given the complexity of the aims, it could benefit from additional clarity More specifically, it would be important to summarise/highlight, after the initial discretion of each WP, what it aims to add to current literature (which are the gaps each WP aims to fill)

We appreciate this generous description of our project. We strive to have a balanced view. We have revised the manuscript to further highlight what the WPs add. Before knowing any results, it is difficult to be very specific about what our study adds to the current literature. Generally, our design enables empirical investigation of research questions related to the well-known variation in ADHD diagnoses and treatment. We believe that much if not most of this variation is due to variation in cultures, attitudes and treatment policies between treatment units (departments, hospitals, regions, etc.), which is caused by a controversy between a restrictive and a liberal position on ADHD diagnosis and treatment. The many RCTs on ADHD cannot solve this dispute, and it is for ethical and practical reasons difficult to settle this controversy by the RCT design, because it would require randomizing patients on the margin of ADHD diagnosis to diagnosis and treatment. The main contribution of this study is the empirical investigation of this question. We have aimed to clarify this in the revised manuscript.

Also, I'd suggest to provide a more "evidence-based, updated" introduction, rather than citing an old and, to some extent, biased (after the initial RCT phase) study as the MTA. the author may want to cite recent evidence synthesis on ADHD medications effects in the short term (eg:

<https://pubmed.ncbi.nlm.nih.gov/30097390/>) and on the pharmacological strategies- in particular behavioural (eg: <https://pubmed.ncbi.nlm.nih.gov/29083042/>) as WP4 is focused on the issue of the long term, the authors may want to cite and discuss, in the discussion section, the main points in a controversy paper in JAACAP (<https://pubmed.ncbi.nlm.nih.gov/31254608/>, <https://pubmed.ncbi.nlm.nih.gov/31515164/>, <https://pubmed.ncbi.nlm.nih.gov/31515165/>), highlighting how their project addresses them

We agree that some of our references were old and outdated and are thus deleted in the revised manuscript. We have also included the recommended additional references, and we have updated the manuscript accordingly, including all the suggested references.

Also, as the field lacks a standard definition of "long-term", the authors should clarify what "long-term" refer to in their project

We believe that it is difficult to give a standard definition of "long-term" in this context in terms of months or years as this would depend on when ADHD is diagnosed, and which type of endpoint this regards. In this context, "long-term" means years rather than months, and all our end-points regard the transition between childhood or adolescence to young adult life. We have specified this in the revised manuscript.

It would also be good to highlight also what their project adds to the large number of studies form the Swedish, USA, and Hong Kong registries (<https://pubmed.ncbi.nlm.nih.gov/31155139/>)

The main contribution of our study is the use of empirical use of causal modelling, and the theoretical focus on the variation in diagnosis and medication. This is highlighted in the revised manuscript.

Reviewer: 2

Reviewer Name: Peik Gustafsson

Institution and Country:

Lund University, Medical Faculty, Department of Clinical Sciences Lund, Child and Adolescent Psychiatry.

Lund, Sweden

Competing interests: None declared

Comments to the Author

Your planned project is very interesting. I have only one concern: You will base your study on differences in provider preference. This is an interesting approach as a kind of quasi-experimental design. The problem with this design is that there may be some differences in the true prevalence of ADHD and in referral frequencies in different geographical areas, not just reflecting provider preference. The problem will be to analyse relevant potential confounding and compensate for it, which might be difficult. Examples of confounding could be that some people might move to a university city for studies and academic career and that fewer parents of children with ADHD will do this (many of these parents have ADHD themselves). Special schools and better chance of getting a job could cause individuals (parents and children) with ADHD to move into a certain area. Different geographical areas might have different knowledge of and different attitudes towards the ADHD diagnosis, causing differences in referral frequency not reflecting the provider preference. I would appreciate if you could discuss these problems in greater detail and make a list of such potential confounders and to what degree you could compensate for them for in your study.

We appreciate this comment, and we agree this is the Achilles heel of our identification strategy (empirical strategy). Since we submitted the protocol, we have actually planned to investigate if the variation in rates of ADHD diagnosis and medication is associated with the observed variation in ADHD symptoms. We will use the Norwegian Mother and Child study for this purpose, which contains the relevant data on ADHD symptomatology, and test the hypothesis on an aggregated geographical variation. A nil finding will support the planned instrument. On the contrary, we will also examine if there is a positive association between clinicians' attitudes and opinions on ADHD diagnosis and medication, and the registry data variation in ADHD diagnosis and medication. A positive association

will strengthen the instrument. These parts of the project were not sufficiently described in the originally submitted manuscript, and are now described in more detail.

VERSION 2 – REVIEW

REVIEWER	Samuele Cortese University of Southampton, UK
REVIEW RETURNED	20-Dec-2020
GENERAL COMMENTS	Thank you for addressing my suggestions. I do have any further comment.
REVIEWER	Peik Gustafsson Lund University, Medical Faculty, Department of Clinical sciences, Lund, Child and adolescent psychiatry. Lund, Sweden
REVIEW RETURNED	09-Dec-2020
GENERAL COMMENTS	I am satisfied with your revised version and with the description of your plans to control for confounders concerning possible geographical differences in the true prevalence of ADHD. I think that your article is ready for publication in its present form.